# Blind estimation and correction of microarray batch effect

**Sudhir Varma** *

HiThru Analytics, HiThru Analytics LLC, Princeton, NJ, United States of America

* sudhir.varma@hithru.com

**Data Availability Statement:** All data are taken from publicly available Gene Expression Omnibus projects (List of projects available in Supporting Information Tables).

**Funding:** The author (SV) does contract statistical analysis under the business name of "HiThru

## Abstract

Microarray batch effect (BE) has been the primary bottleneck for large-scale integration of data from multiple experiments. Current BE correction methods either need known batch identities (*ComBat*) or have the potential to overcorrect, by removing true but unknown biological differences (Surrogate Variable Analysis *SVA*). It is well known that experimental conditions such as array or reagent batches, PCR amplification or ozone levels can affect the measured expression levels; often the direction of perturbation of the measured expression is the same in different datasets. However, there are no BE correction algorithms that attempt to estimate the individual effects of technical differences and use them to correct expression data. In this manuscript, we show that a set of signatures, each of which is a vector the length of the number of probes, calculated on a reference set of microarray samples can predict much of the batch effect in other validation sets. We present a rationale of selecting a reference set of samples designed to estimate technical differences without removing biological differences. Putting both together, we introduce the Batch Effect Signature Correction (*BESC*) algorithm that uses the BES calculated on the reference set to efficiently predict and remove BE. Using two independent validation sets, we show that *BESC* is capable of removing batch effect without removing unknown but true biological differences. Much of the variations due to batch effect is shared between different microarray datasets. That shared information can be used to predict signatures (*i.e.* directions of perturbation) due to batch effect in new datasets. The correction can be precomputed without using the samples to be corrected (blind), done on each sample individually (single sample) and corrects only known technical effects without removing known or unknown biological differences (conservative). Those three characteristics make it ideal for high-throughput correction of samples for a microarray data repository. We also compare the performance of *BESC* to three other batch correction methods: *SVA*, Removing Unwanted Variation (*RUV*) and Hidden Covariates with Prior (*HCP*). An R Package *besc* implementing the algorithm is available from http://explainbio.com.

## Introduction

Batch effect (BE) has been the primary bottleneck for the large-scale integration of data from multiple experiments. BE, defined as the systematic biases between microarray data generated

Analytics LLC". Currently he is working as a contractor, part time with the National Institutes of Health (Bethesda MD) and part time with Tridiuum Inc. (Philadelphia PA). There are no other owners of employees of HiThru Analytics and it is not a subsidiary of any other company. The Funder provided support in the form of salaries for authors (SV), but did not have any additional role in the study design, data collection and analysis, decision to publish, or preparation of the manuscript. The specific roles of these authors are articulated in the 'author contributions' section.

**Competing interests:** The author (SV) does contract statistical analysis under the business name of "HiThru Analytics LLC". Currently he is working as a contractor, part time with the National Institutes of Health (Bethesda MD) and part time with Tridiuum Inc. (Philadelphia PA). This does not alter our adherence to PLOS ONE policies on sharing data and materials.

by different labs at different times or under different experimental conditions [1, 2], can act as a confounding variable in statistical tests and usually has a stronger effect on the measured expression than the biological phenotype under study [3].

Unknown or unrecorded experimental or biological differences can add a systematic difference between putative replicates within or between two batches. Thus, we use the term *batch effect* as a general term for any heterogeneity due to experimental factors between samples that are putative experimental replicates. The heterogeneity can extend to different samples within the same sample collection, *i.e.* considering only average difference between two collections will likely underestimate the BE.

In practice, it has proven difficult to separate heterogeneity due to technical differences from that due to unknown biological differences. The usual approach in batch correction [4–6] is to protect the known covariates and remove all remaining heterogeneity. Biological differences such as sex and genotype can be clinically important but they will be removed if they are not part of the protected covariates [7, 8]. Conversely, if the study design is unbalanced, the statistical significance of the association of gene expression with the protected covariates can be inflated beyond what one would expect by just a reduction in noise [9].

Additionally, current batch correction methods are intended to be used each time a new composite dataset is created. It is known that specific differences in sample condition, experimental technique [1, 10] and environmental conditions [2] can affect the measured gene expression in predictable directions irrespective of the sample type. However, there has been little systematic effort to estimate how many of those common effects are shared between datasets or to compute dataset-independent batch correction parameters that can be used for "blind" prediction of BE.

In this article, we take another approach: instead of estimating and removing all differences between two batches, we only remove those differences that are known to be associated with technical variations. We show that a large proportion of the BE in Affymetrix U133 Plus2 array data can be captured by a relatively small set of signatures, defined as the directions in which the measured expression has been perturbed by batch effects. We estimate *batch effect signatures* (BES) in the form of orthogonal components from a large reference dataset of samples. We develop an algorithm for computing the BES using the reference dataset such that the BES are unlikely to model known or unknown biological differences. We introduce a novel batch-correction method called Batch Effect Signature Correction (*BESC*) that uses the batch effect signatures for blind prediction and correction of BE in new samples and compare the performance to *SVA*.

## Materials and methods

### Batch effect and correction methods

The measured expression for a set of samples can depend on one or more biological factors (such as cell line name, tissue of origin or disease status) and unknown or unmodeled experimental factors (such as microarray batch, FFPE vs. fresh samples, and experimental technique). Following [4] we can model the expression of a sample as a linear combination of known biological covariates, unknown batch effect and noise

$$x_{ij} = \mu_i + f_i(y_j) + \sum_{l=1}^{L} \gamma_{il} g_{lj} + e_{ij} \tag{1}$$

Where $x_{ij}$ is the measured expression of gene $i$ (out of $m$ genes) for sample $j$ (out of $n$ samples), $\mu_i$ is the overall mean expression of gene $i$, $f_i(y_j)$ is a (possibly non-linear) function that

models the dependence of the expression of the $i^{th}$ gene on the $j^{th}$ known biological factors $y_j$. The batch effect is modelled as a linear combination $\sum_{l=1}^{L} \gamma_{il}g_{lj}$ of $L$ experimental factors each composed of $g_{lj}$ the effect of the $l^{th}$ experimental factor on the $i^{th}$ gene multiplied by weight $\gamma_{il}$. The last term $e_{ij}$ is uncorrelated noise.

Eq 1 indicates that the overall space of measured gene expressions can be separated into a space spanned by the biological variation $f_i(y_j)$ and a space spanned by systematic batch effects $\sum_{l=1}^{L} \gamma_{il}g_{lj}$. The purpose of BE correction is to estimate and remove the latter (third term on the RHS of Eq 1) while retaining the biological differences. *Data-specific* batch correction methods (such as *ComBat* and *SVA*) assume that the BE space (third term) is unique to each composite dataset and has to be recalculated each time a new composite dataset is created. However, we show that an orthogonal basis derived from the BE space of a reference dataset can be used to estimate and remove the variation in the BE space for other test datasets (*i.e. blind batch correction*).

**Surrogate Variable Analysis (SVA).** SVA [4] has been widely used for estimating hidden covariates (including technical and biological). Fitting the functions $f_i$ and global means $\mu_i$ using linear or non-linear regression, we can express Eq 1 in terms of the residuals of the fit

$$r_{ij} = x_{ij} - \mu_i - f_i(y_j) = \sum_{l=1}^{L} \gamma_{il}g_{lj} \tag{2}$$

The aim of *SVA* is to find a set of $K \leq L$ orthogonal vectors (*surrogate variables*) that span the same linear space as $g_l$

$$\sum_{k=1}^{K} \lambda_{ik}h_{kj} \approx \sum_{l=1}^{L} \gamma_{il}g_{lj} \tag{3}$$

Each surrogate variable $h_k = [h_{k1}, h_{k2}, \ldots, h_{kn}]^T$ is a vector of length equal to the number of samples that models one hidden covariate that is not present in the known covariates, and $\lambda_{ik}$ is the influence of $h_k$ the measured expression of the $i^{th}$ gene. Together they can be used to model heterogeneity from unknown sources in any future statistical analysis on that dataset. *SVA* proceeds to find these surrogate variables by doing a Singular Value Decomposition of the residuals $r_{ij}$.

The advantage of *SVA* is that we do not need to know the actual batches. The disadvantages of *SVA* are that, firstly, it has to be recomputed on each new dataset, secondly, it can remove unknown but important biological differences between samples.

**Removing Unwanted Variation (RUV).** RUV [11] is a batch correction algorithm that uses factor analysis on control genes (i.e. genes that are not known to be differentially expressed for any of the known covariates) to estimate and remove batch effect. Apart from selection of the control genes, there is a parameter $\nu$ that has to be adapted to the dataset being corrected. RUV can be used without knowing the actual batches, but the selection of the control genes and $\nu$ remain challenging. Furthermore, these two selections are only valid for a particular dataset; they have to be re-evaluated for each new dataset.

**Hidden Covariates with Prior (HCP).** HCP [12] is an approach to estimating batch effects modelled as a linear combination of known covariates. HCP aims to be a generalization of several factorization-based batch correction methods by modelling both the known and unknown biological or technical covariates. The weights given to the known and unknown variation results in three parameters $\lambda, \sigma_1, \sigma_2$ that have to be optimized for each dataset. HCP has the disadvantage that these parameters are specific to a particular dataset and have to be re-computed for each new dataset. Furthermore, without some ground truth to compare to

(e.g. gene co-expression compared to known co-expressions, detection of known eQTLs), it is not possible to find the values for the three parameters that minimize the batch effect.

**Batch Effect Signature Correction (BESC.** We introduce a novel approach *BESC* that aims to learn the variation due to unknown technical differences using a reference dataset and apply it to correct batch effect in other datasets. To motivate our approach, note that $\lambda_{ik}$ is the difference in the expression of the $i^{th}$ gene for each unit change in the $k^{th}$ surrogate variable. Thus, the vector $\lambda_k = [\lambda_{1k}, \lambda_{2k}, ..., \lambda_{mk}]$, for the $k^{th}$ surrogate variable and each genes $1...m$, is a *signature* that quantifies the dependence of the expression of each gene on the $k^{th}$ surrogate variable. For example, if the surrogate variable captures the effect of a particular type of sample preparation, the *signature* is the differential expression between samples that use that preparation method *vs.* samples that do not. We call the set of *K* vectors $\lambda_k$ the *Batch Effect Signatures* (BES). Each vector is a zero-mean unit vector (*i.e.* sum of square of components equals one) that is as long as the number of probes on the array. We can consider batch effect to be a perturbation in a certain direction for all samples in the same batch. These perturbations can be estimated by looking at the expressions of the same cell line in different batches. Each perturbation is a sum of contributions from multiple sources of technical variation. BES decomposes the individual contribution of each source of technical variation by looking for a set of vectors that spans the batch effect perturbation-space.

Our claim is that the expression difference (*i.e.* signature $\lambda_k$) between samples that differ on a certain experimental factor (*e.g.* RNA amplified vs. un-amplified) would be more likely to remain the same, multiplied by some coefficient even if the surrogate variables do not remain the same. Given a dataset of reference samples that is large and diverse enough, we can compute the signatures of the various known or unknown factors that contribute to BE. Those signatures can then be used to estimate and remove BE from new samples. We call this approach the *Batch Effect Signature Correction* (*BESC*).

## Difference between SVA and BESC

There are some differences between the *SVA* and *BESC* formulations. Writing Eq (2) in matrix form

$$R = DU \tag{4}$$

Where *R* is the *m×n* matrix of residuals (*n* samples, *m* genes), *D* is the *m×K* matrix of batch effect signatures ($\lambda_{ik}$ the element at $i^{th}$ row and $k^{th}$ column) and *U* is the *K×n* matrix of surrogate variables ($h_{kj}$ the element at $k^{th}$ row and $j^{th}$ column). *SVA* constrains the rows of *U* to be orthogonal and estimates the matrix *U* using singular value decomposition (SVD) on the residual matrix *R* (which also constrains the columns of D to be orthogonal). On the other hand, we constrain the columns of *D* to be orthogonal and estimate *D* by taking the Principal Components (PCA) of the transpose of the residual matrix, $R^T$. Note that there is no substantial difference between using SVD or PCA for decomposing *R* (except computational stability).

Strictly speaking, *SVA* is not a batch correction algorithm. The surrogate variables also capture heterogeneity due to unknown biological differences since they are calculated without reference to the batch. Other batch correction methods such as *ComBat* [5] require known batch assignment, *i.e.* which samples belongs to which batch. A modification to *SVA*, *permuted-SVA* (or *pSVA*) [8] has been proposed to prevent the algorithm from removing unknown biological differences. However, *pSVA* has the same limitation as ComBat, *i.e.* it is applicable only when the technical covariates that contribute to batch effect are known. As *SVA* does not require that information, it is the closest comparable algorithm to *BESC* and was selected for comparison to *BESC* in this article.

## Selecting samples for the reference set

The selection of the reference set is crucial to ensure that the BES computed from it does not capture unknown biological covariates of the samples (*e.g.* sex, genotype). We selected and annotated a reference set of 2020 cell line samples (242 unique cell lines) on the Affymetrix U133 Plus 2.0 platform, selected from 348 collections from the Gene Expression Omnibus (S1 Table). Fitting the linear model in Eq (1) using the cell line name as the known covariate captures all known and unknown biological differences between the samples. The same cell line, under untreated conditions, should give similar expression profiles between replicates.

Although it is possible for growth conditions (*e.g.* passage number, growth medium) to affect the expression for a cell line, it can be argued that cell lines that have been grown from a standardized population of cells are one of the most replicable biological samples. Any differences in expression measured in two batches can be treated as mostly arising from BEs and experimental noise.

## Correcting BE using BES

Given a set of BES, the batch effect in a new set of samples is computed by fitting a linear model of the BES to the expressions of each sample, *i.e.* we find weights $a_k$ that minimize the squared residuals

$$\left\| x_{new} - \sum_{k=1}^{K} a_k \lambda_k \right\|^2 \tag{5}$$

Where $x_{new}$ is the vector of expressions for a new sample to be corrected and $\lambda_k$ is the $k^{th}$ BES. $\sum_{k=1}^{K} a_k \lambda_k$ is the estimate of the BEs and the residuals $x_{corr} = x_{new} - \sum_{k=1}^{K} a_k \lambda_k$ is the corrected expression for the sample. Since the BES $\lambda_k$ are zero sum orthonormal vectors, it is simple to show that $a_k = \lambda_k^T x_{new}$ where the superscript T indicates the transpose, *i.e.* the weights for each BES is the scalar product of the BES and the expression vector.

BES is thus a "blind" estimate of the BE; we do not recalculate BE parameters on the set of new samples. That enables *single sample correction* of new samples, *i.e.* without having to re-compute the correction for all the samples whenever new samples are added.

## Quantifying batch effect

Several methods have been proposed in literature to visualize or quantify the batch effect in a dataset [13]. Visualization methods include clustering (dendrograms) and principal component analysis (PCA). However, they are not suited for very large number of samples, as is the case for the datasets in this paper. Principal Variance Component Analysis (PVCA) [1] has been proposed as a quantitative measure. PVCA fits a linear mixed model to the principal components of the data using the known biological covariates and batch identities as random effects. The variance due to the different covariates and batch is summed up across the principal components and offers a way to compare the primary sources of variance. For the two validation sets, we used PVCA to compute the contribution of variance due to sample-type as a measure of the batch effect. The higher the contribution, the lower the batch effect.

However, PVCA has the disadvantage that the variances due to batch effect in different test datasets cannot be averaged together (as is required for cross-validation). So, we developed another batch effect measure, the Distance Ratio Score (DRS) that is intuitive, provides one value for each sample and can be averaged across various test sets. Consider a set of samples from one or more sample types hybridized in multiple batches. For a sample of a certain sample type, we take the log of the ratio of the distance to the closest sample of a different sample

type to the distance to the closest sample belonging to a different batch but the same sample type.

$$DRS = \frac{1}{n}\sum_{i=1}^{n}log_2\left(\frac{d(x_i, x_{i,dt})}{d(x_i, x_{i,db/st})}\right) \tag{6}$$

Where $d()$ is any measure of dissimilarity between two samples, $x_i$ is the $i^{th}$ sample, $x_{i,dt}$ is the closest sample of a different sample type and $x_{i,db/st}$ is the closest sample from a different batch of the same sample type as $x_i$. Intuitively, the DRS is high if samples of the same type cluster together irrespective of batch since the denominator will be small compared to the numerator. Conversely if most of the samples cluster according to batch rather than sample type, the DRS will be small. For our analysis, we used the Euclidean distance as the dissimilarity measure.

## Cross-validating batch effect signatures

The question remains whether BES calculated on one dataset are general enough to be predictive of the BE in another dataset. To investigate that, we did 5-fold cross-validation, *i.e.* created 5 random splits of the samples in the reference set into training and testing sets (approximately 80% samples for training and 20% samples for testing) ensuring that all samples from an individual collection are either all in the testing set or in the training set (S1 Table).

For each train/test split, we used the training set to compute the residuals after fitting a model to the expression with the known sample type as the covariate. Then we computed the principal components of the transpose of the residual matrix. The eigenvectors of the covariance matrix (*i.e.* the principal components) were used as putative BES. We used varying numbers of eigenvectors with the highest eigenvalues to correct the samples in the testing set and computed a Distance Ratio Score (DRS) of the test set samples for each number of BES.

## Consistency of BESC corrections for different reference sets

Since we claim that the BESC algorithm picks up batch effects due to common technical differences, it should pick the same corrections when trained on different reference sets. To investigate that, we used the 5 cross-validation splits of the reference set to create two smaller reference sets: one reference set using splits 1 and 2 together and another reference set using splits 3 and 4 together. From each reference set, we computed a set of BES: $BES_{1+2}$ from splits 1 and 2 and $BES_{3+4}$ from splits 3 and 4. Using these two sets of BES, we did three analyses. First, both sets of BES were used to correct the $5^{th}$ split with different numbers of BES. We visualized the two sets of corrected data for split 5 using principal component analysis (PCA). Second, we corrected random vectors with the two BES and computed the correlations between the corrections for different numbers of BES. Third, we corrected the colon cancer dataset (validation set 2) with both sets of BES and compared the list of genes differentially expressed between MSI and MSS samples. Results for these analyses are in S1 File.

## Testing on validation sets

We selected two validation sets to test the effectiveness of the BES calculated on the reference set for removing BE. All of these samples were collected from public GEO datasets. Annotations for the samples were compiled from annotations provided by the original data submitter.

Different numbers of the top BES were used to estimate and remove the BE in the validation sets and the DRS BE score was calculated. Note that the BES were calculated using only the reference set samples. The sample type (or any other covariate) of the validation set samples were not used during the BE correction.

**Validation set 1 (Primary normal samples)**– 878 samples of primary healthy tissue (532 blood, 228 colon and 118 lung samples from 41 collections) on the U133 Plus 2.0 platform from GEO (S2 Table). We predicted the sex of each sample using 5 chromosome-Y genes (S1 File). Samples were selected to create a balanced dataset with respect to collection, organ and sex.

**Validation set 2 (Colon cancer and normal)**– 3041 samples of primary colon cancer and normal samples (476 normal colon and 2565 colon cancer samples from 60 GEO collections) on the U133 Plus 2.0 platform from GEO (S3 Table). We have compiled the reported Micro-Satellite Stability/Instability (MSS/MSI) status for 513 out of the 2565 colon cancer samples (154 MSI, 359 MSS). Out of those 513, we selected a balanced set of 503 samples with known MSI/MSS status to test for differential expression between MSI and MSS.

### Permutation p-value for DRS

To test whether the improvement in the DRS is statistically significant, we performed a permutation test of the DRS obtained with various numbers of BES. We permuted the data for each gene in the reference set to destroy any batch effect signatures and then computed the BES using that null dataset. Varying numbers of those null BES were used to correct the samples in the validation set and the DRS obtained was compared to the true DRS at different numbers of BES. The procedure was repeated 100 times and the null DRSs used to compute the p-value for the true DRS using a Student's t-test at each number of BES.

### Comparison to SVA, RUV and HCP

We compared the DRS for the validation set for increasing numbers of BES to that for increasing levels of correction for *SVA*, *RUV* and *HCP*. The comparison was done for varying number of 1) surrogate variables (*SVA*), 2) factors (*RUV*) and 3) estimated hidden covariates (*HCP*). Note that it is not a direct comparison, since the other three methods use the validation set samples to compute the BE parameters. The training set/validation set approach we have taken with BES cannot be applied to *SVA*, *RUV* or *HCP* since they are expected to be run on the same dataset that is being corrected.

For *RUV*, we selected the set of housekeeping genes as well as the value of $\nu$ that gave the best performance in terms of DRS and PVCA (S1 File). For *HCP*, we did a grid search over the three parameters $\lambda, \sigma_1, \sigma_2$ and selected the values that gave the highest DRS.

One disadvantage of *SVA* is that it will remove unknown biological differences (*e.g.* sex) from the cleaned data [7, 8]. To show that, we predicted the sex of each sample in the validation set 1 using the expression of chromosome Y genes (RPS4Y1, KDM5D, USP9Y, DDX3Y, EIF1AY, see S1 File) and looked at the number of genes significantly different between male and female samples at various levels of correction for *BESC* and *SVA*. For validation set 2, we had the reported MSI status for 513 colon cancer samples. We looked at the number of genes significantly different between MSI and MSS samples. The computation of statistical significance for validation set 1 was done using organ source, sex and batch as covariates and for validation set 2 was done using batch and MSI/MSS status as covariates.

## Results

### BES computed on training data is predictive of batch effect in test data

Fig 1 shows the 5-split cross-validated batch effect DRS on the reference set. The plot shows the average DRS on the test set corrected using BES calculated on the training set. The DRS increases as the number of BES increases, indicating reduction of BE with increasing

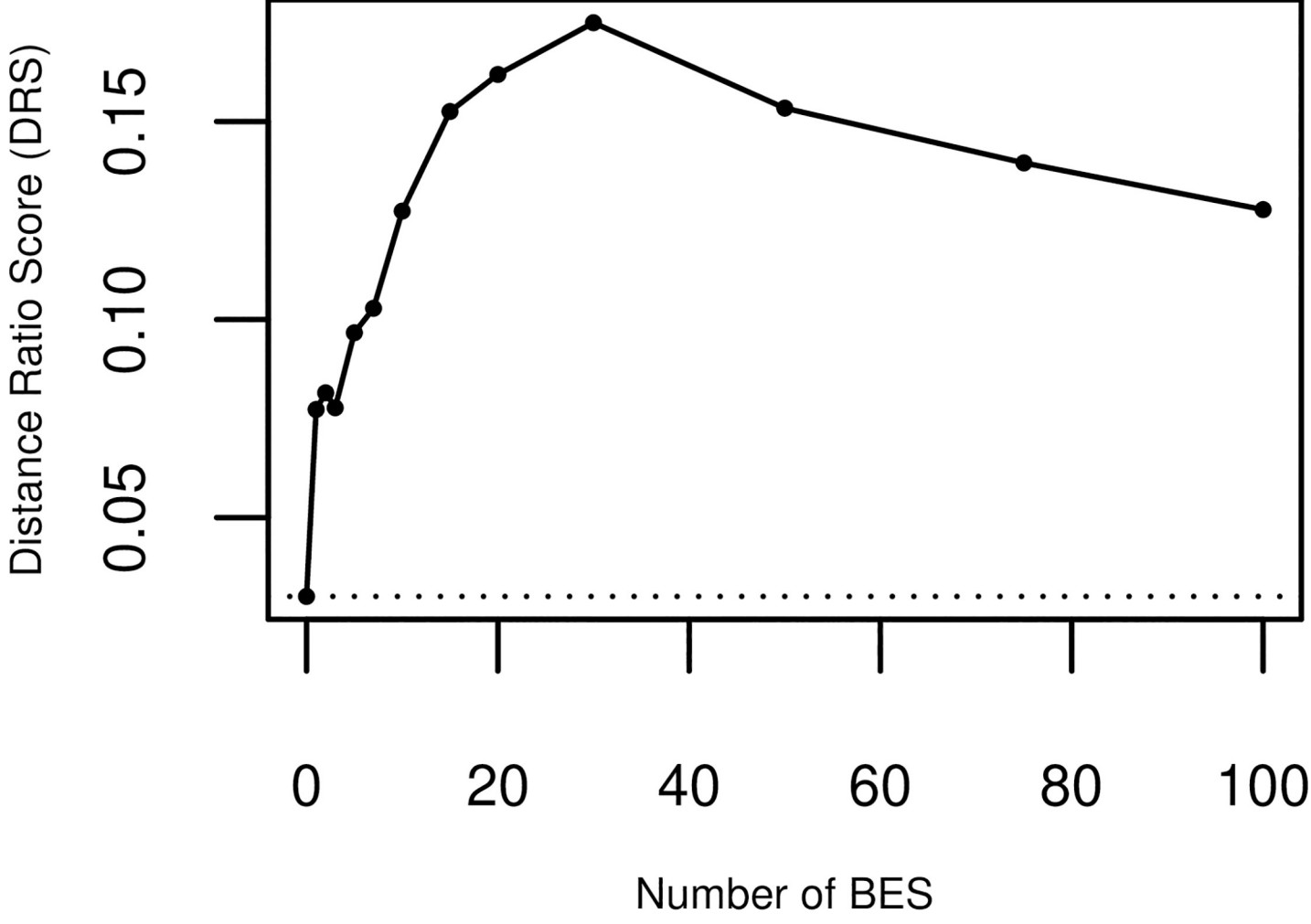

**Fig 1. Cross-validated performance on reference set.** The cross-validated Distance Ratio Score (DRS) for the reference set vs. the number of Batch Effect Signatures (BES) used for the correction. Higher DRS indicate lower levels of batch effect. The DRS reaches a maximum for 30 BES.

correction. It reaches a maximum with 30 BES used in the correction, indicating that the first 30 Batch Effect Signatures estimated on the training set capture variation due to BE in the test set. Further correction reduces the DRS, indicating that the BES above 30 do not capture any information about the BE.

## BES computed on different reference sets are consistent

S5 Fig shows the results of applying BESC using BES computed from different reference sets. As long as the number of BES is smaller than 10, the corrected samples from the two sets of BES stay close together (while moving away from the uncorrected samples). After 10 BES, the two sets of corrected samples start separating. S6 Fig shows the mean correlation between the corrections performed by the two sets of BES on a random vector. The correlation increases with increasing number of BES, reaching a maximum at 10 BES and then goes down. Furthermore, S7 Fig shows the overlap percentage between genes differentially expressed between MSI and MSS for validation set 2 corrected by the two sets of BES. The overlap remains high (>80%) for up to 10 BES.

Together, the three results show that two reference sets (splits 1+2 and splits 3+4), pick up similar batch effect signatures (until ~ 10 BES) indicating that the algorithm is picking up consistent correction factors from different reference sets.

## BES computed on reference set corrects batch effect in validation sets

Fig 2A shows the DRS on validation set 1 using various numbers of BES calculated on the reference set. The figure compares the performance of the *BESC* method to the *SVA*, *RUV* and *HCP* methods using the same number of correction factors as the number of BES. The *BESC* method shows improvement in the DRS when up-to 15 BES are used for correction. Using more than 15 signatures begins to decrease the effectiveness. *SVA* shows a steady improvement in batch effect at each level while *RUV* shows improvement up to two correction factors and then saturates. *HCP* shows improvement for the first correction factor, but its performance decreases once 20 or more correction factors are included. Fig 2B shows similar results using the variance contributed to the sample type (in this case, the organ of origin of the samples). The variance increases for the sample type (with a corresponding decreasing in variance due to batch effect; not shown) as the number of BES or number of surrogate variables increases. As with the DRS (Fig 2A), *SVA* show a steady improvement in batch effect at each level. The performance for *BESC* reaches a maximum when 15 BES are used. *RUV* and *HCP* do not show any improvement with correction (with the performance of *HCP* decreasing over the un-corrected data).

Fig 3A shows the DRS on validation set 2 for varying number of BES used in the correction. For that dataset too, the DRS reaches a maximum plateau at 15 BES. Using more than 15 BES does not significantly change the DRS. Fig 3B shows similar results using the PVCA-computed variance contribution due to sample-type (*i.e.* disease status- colon cancer *vs.* normal in this case). In both cases, the *SVA* corrected data shows superior performance in removing batch effect.

## Comparison to SVA

*SVA* shows superior performance (Fig 2A) over most of the range of the x-axis (number of BES/number of surrogate variables used for correction).

However, Figs 2C and 3C illustrates the primary disadvantage of *SVA* which is the "normalizing away" of unknown but true biological differences. When sample sex is not included as one of the "protected" covariates in the *SVA* algorithm, the difference between male and female samples (*i.e.* the number of statistically significant genes) decreases with increasing *SVA* correction. On the other hand, the difference is maintained (and slightly increased) with increasing BES correction.

Fig 3C shows the number of genes that are differentially expressed between MSI and MSS samples at a statistically significant level (FDR<0.05). There is a steep drop off of the number of genes for the *SVA*-corrected data. *SVA* correction eventually removes almost all of the differences between the MSI and MSS samples. On the other hand, the BES corrected data retains most of the differential expression.

Figs 2C and 3C emphasize the conservative nature of *BESC*; only differences that are known to be due to BE are removed. Any unknown/unmodeled difference between the samples that is due to true biology is maintained. In contrast, *SVA* captures those differences unless they are protected, and correction with the surrogate variables removes those differences from the samples.

## Comparison to RUV

For validation set 1, *RUV* shows some promise as a batch correction method for small numbers of correction factors (2 or less). However, it fails to do any correction for validation set 2.

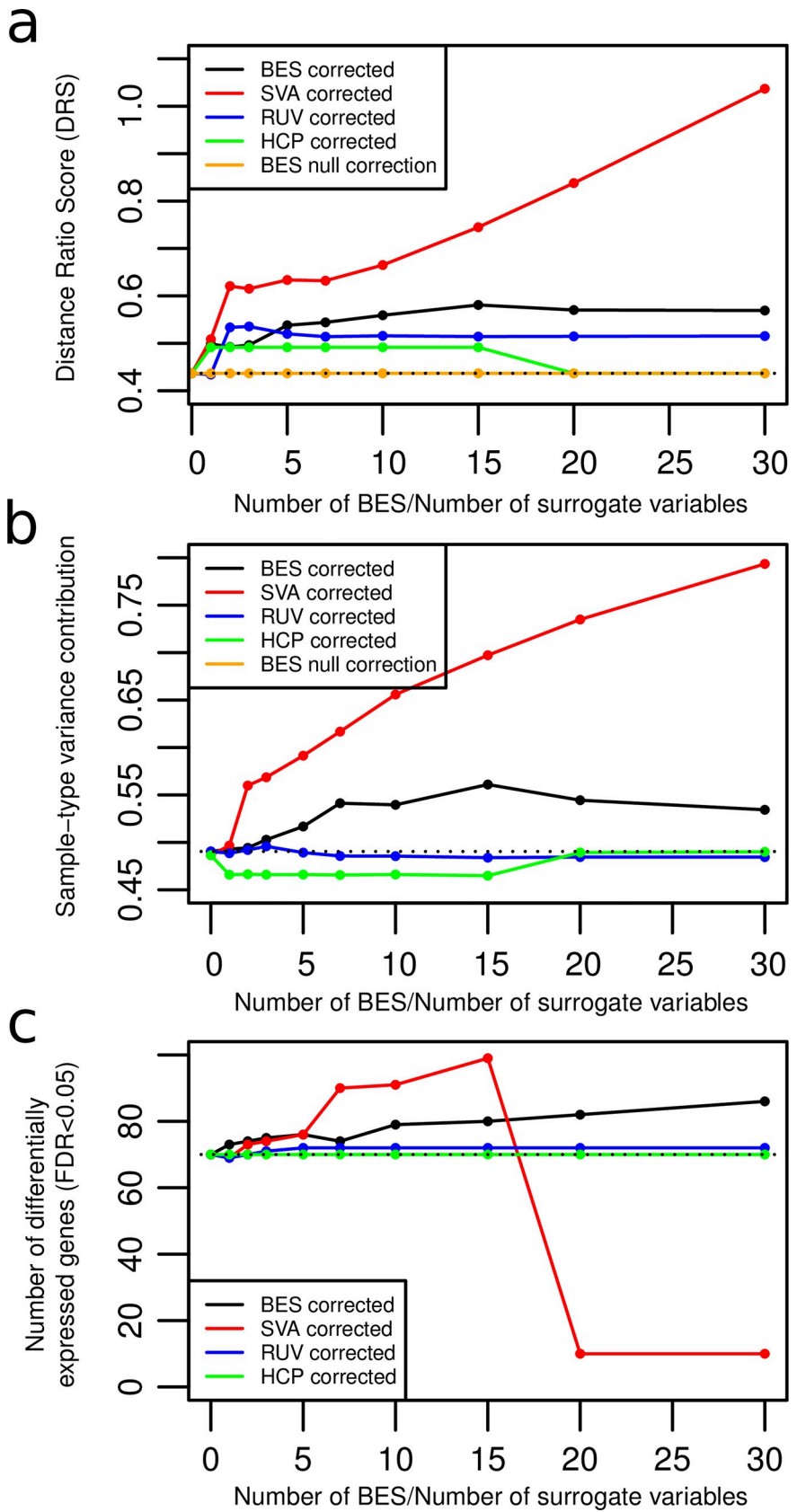

**Fig 2. Performance on validation set 1.** a) DRS for the validation set 1 using *BESC*, *SVA*, *RUV* and *HCP* and the permuted null BES b) Contribution of variance due to organ-type, calculated using PVCA c) Number of genes differentially ex-pressed between male and female samples at various levels of correction.

Also note that the results for *RUV* are selected from that value of *v* and set of housekeeping genes that has the highest performance. In practice, it is computationally cumbersome to select the best parameter.

## Comparison to HCP

*HCP* does some significant correction for validation set 2 (Fig 3) but shows disappointing performance on validation set 1 (Fig 2). In addition, *HCP* had the disadvantage that its performance had to be tuned by optimizing over a grid of three parameters (we show the results for the best performing combination).

## BES correction is statistically significant

The orange line (Figs 2A and 3A) shows the average DRS for the validation sets corrected using the BES calculated on the permuted reference set data. As expected, there is no significant correction of the data (*i.e.* the DRS does not improve from baseline). The z-score p-value of the DRS using the true reference set (black line) is <1e-16 over the entire range of numbers of BES.

## Discussion

We show that the batch effects between different datasets occupy a space that can be characterized by a small set of orthogonal basis vectors. That enables us to compute Batch Effect Signature (BES) vectors that capture the direction of perturbation using a reference dataset and apply them to predict and remove the batch in independent validation datasets. As far as we know, no other "blind" methods of batch correction have been published. All methods, (including *SVA* [4], *RUV* [11] and *HCP* [12]) require the correction factors to be computed on the entire sample set, needing re-calculation each time new samples are added.

Crucial to the correct operation of our algorithm is the selection of a reference set composed of cell lines since specifying the cell line name completely fixes all known and unknown biological covariates. We argue that cell lines that have been grown from a standardized population of cells are one of the most replicable biological experiments, and any difference between the same cell line sample from different experiments is quite likely due only to technical variation. However, note that we can use any sample that has been analysed in multiple experiments by different labs. For example, samples from The Cancer Genome Atlas (TCGA) or any other sample repository have the property that specifying the sample id completely specifies all of the known and unknown biological covariates. All that is required for inclusion in the reference set is that the sample must be uniquely identified across multiple experiments.

We compared the *BESC* algorithm to Surrogate Variable Analysis (*SVA*), Removing Unwanted Variation (*RUV*) and Hidden Covariates with Prior (*HCP*). At larger number of surrogate variables, *SVA* is more effective at detecting and removing residual structure from the dataset, however it is not possible to know how much of that residual structure is due to batch effect compared to unknown but true biological differences. We show that *SVA* is very likely to remove unknown but important biological information in our validation sets (*e.g.* sex in the case of normal samples and MSI *vs.* MSS differences in the case of colon cancer). We show that *BESC* is much more conservative about retaining unknown biological differences

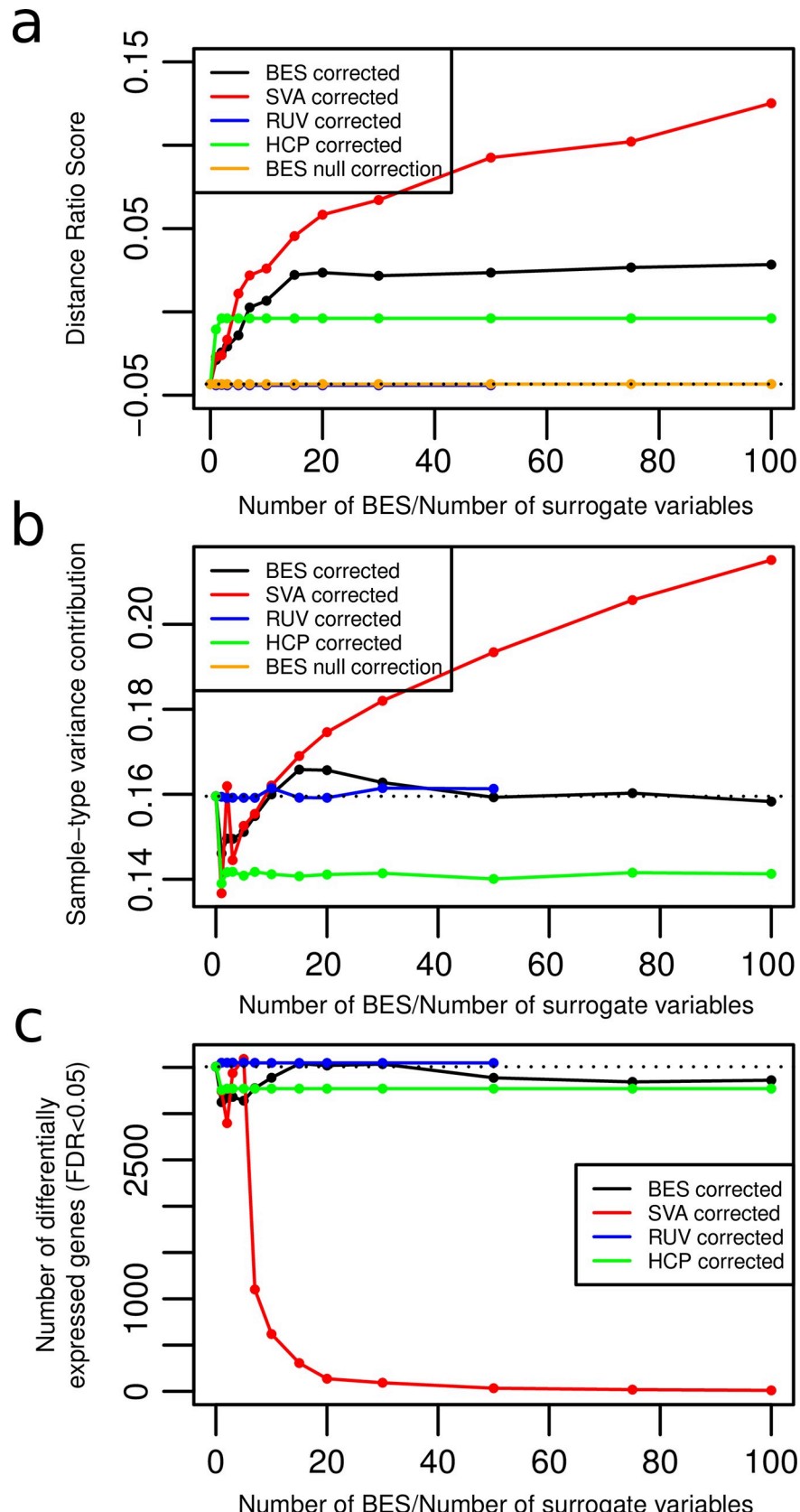

**Fig 3. Performance on validation set 2.** a) DRS for the validation set 2 using *BESC*, *SVA*, *RUV* and *HCP* and the permuted null BES b) Contribution of variance due to disease status, calculated using PVCA c) Number of genes differentially expressed between MSI and MSS samples at various levels of correction by *BESC*, *SVA*, *RUV* and *HCP*.

while removing technical differences. *RUV* and *HCP* show some performance on one validation set each, but do not do any significant correction in the other validation set. Furthermore, both algorithms need to be tuned by testing out various values of tuning parameter(s); a computationally expensive process.

The characteristics of the *BESC* algorithm make it ideally suited for large-scale batch correction in microarray data repositories. *BESC* is conservative, *i.e.* only BE known to be likely due to technical differences is removed. We have shown that biological variation, even if unknown to the *BESC* algorithm, is preserved. *BESC* can be applied to individual samples and does not need to be recomputed as more samples are added to the repository.

One primary disadvantage of the *BESC* algorithm is that it will remove all differences between samples that are parallel to the BES vectors. It is possible that there is important biological information along those directions. However, that biological difference is confounded with the technical differences and cannot be separated out in any analysis.

Another disadvantage is that the current BES vectors are only applicable to the Affymetrix U133 Plus2 array. We used samples from the U133 Plus2 platform to calculate the BES, mainly because it is the most commonly used platform with a large number of cell line samples. Before BES can be used on another platform, we will have to compile a reference set of cell line or other identified samples on that platform and re-compute the vectors on that set. Other BE correction methods are platform agnostic.

## Conclusions

This paper describes a novel finding that batch effect (BE) perturbs measured gene expression in predictable directions which we call Batch Effect Signatures (BES). That characteristic can be used to compute possible directions of the perturbations in a reference dataset which can be used to predict the BE in an independent validation set. Selecting the reference set to contain only known cell-lines ensures that all (known or unknown) biological differences are fixed by specifying the cell-line name. That ensures that the BES capture differences due only to technical differences between the batches.

We show that the BES calculated on the reference set efficiently removes batch effect in two validation sets, as measured by PVCA and the Distance Ratio Score (DRS), a novel measure of batch effect. Compared to *SVA*, our algorithm does not remove all possible differences between samples of the same type in different batches, but we show that *SVA* also over-corrects by removing unknown but true biological differences. Compared to *RUV* and *HCP*, our algorithm shows superior performance over a wider range of datasets.

An R Package *besc* implementing the algorithm is available from http://www.explainbio.com. All data used are public data available from GEO (Gene Expression Omnibus https://www.ncbi.nlm.nih.gov/geo/). GEO accession numbers for all samples used in the analyses are in S1–S3 Tables.

## Supporting information

**S1 Fig. Normalization of arrays.** a) Bias in measured expression that depends on intensity b) Bias in measured expression that depends on array y-axis.
(TIF)

**S2 Fig. Prediction of sex.** Scatterplots of sex related genes with known sex and ellipse of 95% density for fitted Gaussian mixture model.
(TIF)

**S3 Fig. *RUV* parameter selection for validation set 1.** Plot of DRS and PVCA for various levels of the tuning parameter for *RUV* as well as two different sets of housekeeping genes.
(TIFF)

**S4 Fig. *RUV* parameter selection for validation set 2.** Plot of DRS and PVCA for various levels of the tuning parameter for *RUV* as well as two different sets of housekeeping genes.
(TIFF)

**S5 Fig. Consistency of calculated BES.** Overlap between significant genes selected for MSS/MSI differences in validation set 2 for BES calculated on different subsets of the reference set.
(TIFF)

**S6 Fig. PCA of corrected samples for two sets of BES.** Principal Component Analysis (PCA) plots of uncorrected samples and samples corrected using two different sets of BES calculated on different subsets of the reference set.
(TIFF)

**S7 Fig. Correlation between corrected samples.** Correlation between corrected samples at various levels of correction for BES calculated on different subsets of the reference set.
(TIFF)

**S1 Table. List of samples in the reference set (Cell lines) with cross-validation split IDs.**
(XLS)

**S2 Table. List of samples in validation set 1 (Primary normal samples) with predicted sex.**
(XLS)

**S3 Table. List of samples in validation set 2 (Colon cancer and normal) with known MSI/MSS status.**
(XLS)

**S1 File. Supplementary methods.**
(DOCX)

## Acknowledgments

The author gratefully acknowledges the valuable feedback on this paper given by Vinodh Rajpakse and Augustin Luna. The author is also grateful for the constructive suggestions from the two anonymous reviewers which has resolved several errors and made the manuscript stronger overall.

## Author Contributions

**Conceptualization:** Sudhir Varma.

**Data curation:** Sudhir Varma.

**Formal analysis:** Sudhir Varma.

**Investigation:** Sudhir Varma.

**Methodology:** Sudhir Varma.

**Software:** Sudhir Varma.

**Validation:** Sudhir Varma.

**Visualization:** Sudhir Varma.

**Writing – original draft:** Sudhir Varma.

**Writing – review & editing:** Sudhir Varma.

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
