## [Decision Letter · Decision Letter 0]

2 Sep 2019

PONE-D-19-17279

Blind estimation and correction of microarray batch effect

PLOS ONE

Dear Dr. Varma,

Thank you for submitting your manuscript to PLOS ONE. After careful consideration, we feel that it has merit but does not fully meet PLOS ONE’s publication criteria as it currently stands. Therefore, we invite you to submit a revised version of the manuscript that addresses the points raised during the review process.

The manuscript has been assessed by two reviewers; their comments are available below.

The reviewers have raised some major concerns that need attention in a revision. The reviewers note that further information is needed on where the method can be publicly accessed, and they also raise the need to complete further tests as well as comparison to other approaches in order to validate the proposed methodology.

Could you please revise the manuscript to carefully address the concerns raised by the reviewers?

We would appreciate receiving your revised manuscript by Oct 15 2019 11:59PM. To enhance the reproducibility of your results, we recommend that if applicable you deposit your laboratory protocols in protocols.io, where a protocol can be assigned its own identifier (DOI) such that it can be cited independently in the future. For instructions see: http://journals.plos.org/plosone/s/submission-guidelines#loc-laboratory-protocols

We look forward to receiving your revised manuscript.

Kind regards,

Iratxe Puebla

Senior Managing Editor, PLOS ONE

Journal Requirements:

1. Thank you for including your competing interests statement; "The author has declared that no competing interests exist."

We note that one or more of the authors are employed by a commercial company: HiThru Analytics LLC

Reviewers' comments:

Reviewer's Responses to Questions

**Comments to the Author**

1. Is the manuscript technically sound, and do the data support the conclusions?

Reviewer #1: Yes

Reviewer #2: Partly

2. Has the statistical analysis been performed appropriately and rigorously? 

Reviewer #1: Yes

Reviewer #2: Yes

3. Have the authors made all data underlying the findings in their manuscript fully available?

Reviewer #1: Yes

Reviewer #2: Yes

4. Is the manuscript presented in an intelligible fashion and written in standard English?

Reviewer #1: Yes

Reviewer #2: Yes

5. Review Comments to the Author

Reviewer #1: This manuscript presents a novel computational method (BESC) to correct for batch effects in microarray experiments. The novel idea of the author is that he argues that scanning a large representative set of data should be informative enough for parametrizing the "general" batch effects that are inherent to a particular microarray chip once and for all.

If this works, the author points out that one could run BESC once and apply it to new data sets in the future without having to rerun BESC.

I find this a novel, appealing idea. I am not aware of a comparable approach.

The manuscript is well written, clearly understandable and structured.

Major points

(1) The manuscript states that the method is available from http://explainbio.com

But I did not find the software there when I accessed that site.

Hence, I/we could not test the tool ourselves, which is standard when announcing a new bioinformatics tool.

(2) The presented results (figures + suppl figures) appear convincing.

To illustrate the generality and limitations of BESC I suggest to add the following 2 tests:

(a) In addition to the 5-fold CV already done, I suggest that the same 5 data splits should be processed as follows:

-> First, BESC should be parametrized on sets 1 + 2 and then applied to set 5.

-> Alternatively, BESC should be parametrized on sets 3 + 4 and then applied to set 5.

It this manner, two halves of the training data would be used to predict batch-effect corrected versions of set 5. I am curious how different are the results of this exercise.

If one does this for the TCGA data (Fig. 3 C), how large is the overlap coefficient of the differentially expressed genes between the two runs?

(b) The author argued he used data for a large panel of representative cell lines so that the parametrized BESC should be of general nature.

I wonder what the limitations of this approach are?

Does this also work for tissues and conditions that are not represented in the training data set?

This should be demonstrated on a suitable example.

(3) Another research group told us about their experience with TCGA data from the Illumina 27k CpG methylation chip (breast cancer). They found a strong batch effect between separate charges of the chip produced at different times.

Such a case would not be covered by BESC. If you parametrized on MA data collected up to a certain moment in time and apply this to process "all" future data processed with this chip type, one would have to assume that the chip is homogeneously manufactured over many years.

But this may not be the case, and may remain undetected if users simply rely on BESC.

Discussion of this and related effects should be added in the discussion section.

Minor points

(4) throughtout the text many nouns start with capital letters, but need not to do so.

(5) line 317: is to much more -> is much more

(6) line 337: which can used -> which can be used

Reviewer #2: The paper presents a novel method using signatures of batch effects generated from an external reference to eliminate batch effects from microarray data without harming the biological signal.

The idea and method is very interesting and well presented.

1. My main criticism is regarding the validation of the method. the method was compared to SVA which removes batch effects as well as the biological signal. The current validation might just show that SVA removes also biology with the batch effects which is worse than the present method or even the raw uncorrected data. Another comparison should be used for validation. For example, comparison with Combat (RUV etc) or other method that deals with removing batch effects without the biological signal, should be added. It might be also compared to the raw data, showing that it outperform the raw data. In case the new method doesn't correct the data, still it will outperform SVA which overfits the data and eliminated the DEGs.

2. Im missing a presentation of other recent methods that deal with removing batch effects without removing the biological signal. For example, HCP by Mustafavi, BCeF by Somekh etc.

Minor issues:

p.1. line 9 "Even though the effects..." - unclear sentence

p.1. line 22 - very long sentence with too many parentheses.

p.2. l 34 - "or biological" - should be just experimental here?

p.4 - some mistake for gij on the jth gene? not enough explanation on gamma and the factors.

p 5 - need to add more explanation on the SVA formulas. so (3) is currently not fully clear. I had to read the SVA paper to understand.

p 5 l.90 - sometimes ki and sometimes ik are used.

p 5 l.92 - should be "each gene" instead of "all genes"?

l 96 - length one is unclear

l 98 - refine the English sentence

l 102 - please add: to remain the same, multiplied by some coefficient.

p 12 line 246 - Figure not Fig. Figure 2A - please add explanation on what can be seen for SVA.

p. 15 line 294 - unclear sentence

line 299 - for SVA, please add reference

6. PLOS authors have the option to publish the peer review history of their article (what does this mean?). If published, this will include your full peer review and any attached files.

Reviewer #1: No

Reviewer #2: No

---

## [Author Response · Author response to Decision Letter 0]

6 Dec 2019

Response to reviewers

Firstly, I would like to thank the reviewers for the thorough review and very constructive suggestions. I have taken their suggestions to heart and performed extensive work to do the comparisons to other batch effect methods as well as validation methods. I believe this makes the manuscript stronger and improves the presentation considerably.

I have added my responses below in light orange font.

Reviewer #1: 

This manuscript presents a novel computational method (BESC) to correct for batch effects in microarray experiments. The novel idea of the author is that he argues that scanning a large representative set of data should be informative enough for parametrizing the "general" batch effects that are inherent to a particular microarray chip once and for all.

If this works, the author points out that one could run BESC once and apply it to new data sets in the future without having to rerun BESC.

I find this a novel, appealing idea. I am not aware of a comparable approach.

The manuscript is well written, clearly understandable and structured.

Major points

(1) The manuscript states that the method is available from http://explainbio.com

But I did not find the software there when I accessed that site.

Hence, I/we could not test the tool ourselves, which is standard when announcing a new bioinformatics tool.

That was due to a misconfiguration of the web server. The package should be available to download now, on clicking the “BESC R package” link on the top right-hand side. Please make sure to clear your browser cache before reloading http://explainbio.com

(2) The presented results (figures + suppl figures) appear convincing.

To illustrate the generality and limitations of BESC I suggest to add the following 2 tests:

(a) In addition to the 5-fold CV already done, I suggest that the same 5 data splits should be processed as follows:

-> First, BESC should be parametrized on sets 1 + 2 and then applied to set 5.

-> Alternatively, BESC should be parametrized on sets 3 + 4 and then applied to set 5.

It this manner, two halves of the training data would be used to predict batch-effect corrected versions of set 5. I am curious how different are the results of this exercise.

I’ve included a plot that explores this (Supplementary Figure 6). I’ve plotted the uncorrected samples from split 5 along with the same samples corrected by batch effect signatures (BES) calculated on splits 1+2 and splits 3+4 (i.e. two sets of BES). As long as the number of BES is smaller than 10, the corrected samples from the two sets of BES stay close together (while moving away from the uncorrected samples). After 10 BES, the two sets of corrected samples start separating.

Furthermore, Supplementary Figure 7 shows the mean correlation between the corrections performed by the sets of BES computed on splits 1+2 vs 3+4. The correlation increases with increasing number of BES, reaching a maximum at 10 BES and then goes down.

Together, the two results show that two sets of reference data splits (splits 1+2 and splits 3+4), pick up similar batch effect signatures (until ~ 10 BES) indicating that the algorithm is detecting a robust signal.

It seems reasonable that we can find 10 robust BES from a bit less than half the reference data (splits 1+2 or split 3+4) compared to ~20 robust BES from the full reference set. It is possible that additional data in the reference set can give us a larger number of BES covering more of the batch effect 

If one does this for the TCGA data (Fig. 3 C), how large is the overlap coefficient of the differentially expressed genes between the two runs?

I’ve included a plot of the fraction of common genes between the colon cancer/normal dataset (validation set 2) when corrected by sets of BES calculated on the 1st + 2nd split combined and the 3rd + 4th split combined (Supplementary Figure 5). There is a high degree of overlap (>80% intersection) till the number of BES reaches 10. The BES start to diverge from 20 onwards and the overlap decreases.

(b) The author argued he used data for a large panel of representative cell lines so that the parametrized BESC should be of general nature.

I wonder what the limitations of this approach are?

Does this also work for tissues and conditions that are not represented in the training data set?

This should be demonstrated on a suitable example.

The two validation sets are actually different tissues and conditions compared to the reference set that was used to compute the BES. The reference set contains only cell line samples while the validation sets contain primary tissue samples (blood, lung and colon normal samples for validation set 1 and primary colon normal and cancer for validation set 2). 

(3) Another research group told us about their experience with TCGA data from the Illumina 27k CpG methylation chip (breast cancer). They found a strong batch effect between separate charges of the chip produced at different times.

Such a case would not be covered by BESC. If you parametrized on MA data collected up to a certain moment in time and apply this to process "all" future data processed with this chip type, one would have to assume that the chip is homogeneously manufactured over many years.

But this may not be the case, and may remain undetected if users simply rely on BESC.

Discussion of this and related effects should be added in the discussion section.

I do detect a year-dependent effect of one of the BES. The coefficient calculated for samples with different scan years varies smoothly from year to year. It is true that any abrupt large change in the array manufacturing would not be detected by BESC until we have a number of reference samples on the new array. I’ve added a remark in the discussion on this issue.

Minor points

(4) throughtout the text many nouns start with capital letters, but need not to do so.

(5) line 317: is to much more -> is much more Corrected line 372

(6) line 337: which can used -> which can be used Corrected line 395

Reviewer #2: 

The paper presents a novel method using signatures of batch effects generated from an external reference to eliminate batch effects from microarray data without harming the biological signal.

The idea and method is very interesting and well presented.

Major issues:

1. My main criticism is regarding the validation of the method. the method was compared to SVA which removes batch effects as well as the biological signal. The current validation might just show that SVA removes also biology with the batch effects which is worse than the present method or even the raw uncorrected data. Another comparison should be used for validation. For example, comparison with Combat (RUV etc) or other method that deals with removing batch effects without the biological signal, should be added. It might be also compared to the raw data, showing that it outperform the raw data. In case the new method doesn't correct the data, still it will outperform SVA which overfits the data and eliminated the DEGs.

Firstly, I would like to clarify that the comparison to the raw data is present in each figure (the very first point on the left with BES=0, indicating uncorrected data). I’ve added a dotted horizontal line in each figure to show the value for the uncorrected data.

I’ve added comparisons to the results using RUV [1] and HCP [2]. I find that RUV does outperform BESC in validation set 2 for correction with a small number of factors (Figure 3). However, this best performance is selected after choosing between two different sets of control genes and varying the value of ν (a tunable parameter for RUV). In contrast BESC (and SVA) does not need to be tuned. 

The results with HCP was disappointing for both validation sets (Figures 2 and 3). In both cases HCP does not do any better than the raw data. Note that these are the best results as selected from a range of values for the three HCP parameters λ,σ_1,σ_2.

The tuning of the parameters for RUV and HCP are described in the Supplementary Methods.

2. Im missing a presentation of other recent methods that deal with removing batch effects without removing the biological signal. For example, HCP by Mustafavi, BCeF by Somekh etc.

I’ve added comparisons to RUV [1] and HCP [2] in the manuscript and discussion of the results from those algorithms. BCeF [3] appears to be a framework to evaluate and compare different batch correction methods based on how closely they recapitulate a known gene co-expression matrix. They report that none of the batch correction methods is able to fully recapitulate all the unmodelled confounders. As I understand it, they do not propose any new batch correction methods.

1. Gagnon-Bartsch JA, Speed TP. Using control genes to correct for unwanted variation in microarray data. Biostatistics. 2012;13: 539–552. doi:10.1093/biostatistics/kxr034

2. Mostafavi S, Battle A, Zhu X, Urban AE, Levinson D, Montgomery SB, et al. Normalizing RNA-Sequencing Data by Modeling Hidden Covariates with Prior Knowledge. PLOS ONE. 2013;8: e68141. doi:10.1371/journal.pone.0068141

3. Somekh J, Shen-Orr SS, Kohane IS. Batch correction evaluation framework using a-priori gene-gene associations: applied to the GTEx dataset. BMC Bioinformatics. 2019;20: 268. doi:10.1186/s12859-019-2855-9

Minor issues:

p.1. line 9 "Even though the effects..." - unclear sentence I’ve rewritten the sentence (p.1. line 9)

p.1. line 22 - very long sentence with too many parentheses. I’ve rewritten the sentence (p.1. line 23)

p.2. l 34 - "or biological" - should be just experimental here? I have tried to differentiate between batch effect caused due to unknown technical effects from that due to biological effects (in p.2. line 37). Most batch correction methods (e.g. ComBat) remove both, but BESC attempts to remove only the technical differences.

p.4 - some mistake for gij on the jth gene? not enough explanation on gamma and the factors. I’ve explained gamma and tried to make the batch effect term clearer. glj (note it is “l”, a lower case “L”) is the effect of the batch l on sample j. The dependence on the gene “i” is modeled by γ_"il" . (p.4. line 77)

p 5 - need to add more explanation on the SVA formulas. so (3) is currently not fully clear. I had to read the SVA paper to understand. I’ve expanded the description (page 5)

p 5 l.90 - sometimes ki and sometimes ik are used. Corrected (p.6. line 122)

p 5 l.92 - should be "each gene" instead of "all genes"? Corrected (p.6. line 124)

l 96 - length one is unclear I’ve made it clearer (p.6. line 128)

l 98 - refine the English sentence I’ve re-written the sentence to make it clearer (p.7. 129-134)

l 102 - please add: to remain the same, multiplied by some coefficient. I’ve added that phrase (p.7. line 137)

p 12 line 246 - Figure not Fig. Figure 2A - please add explanation on what can be seen for SVA. I’ve updated the figure with results for RUV and HCA with description of the results for each algorithm (p.14. line 290)

p. 15 line 294 - unclear sentence I’ve re-written the sentence (p.16. line 349)

line 299 - for SVA, please add reference Added for SVA, RUV and HCP (p.17. line 353)

---

## [Editor Report · Decision Letter 1]

16 Jan 2020

PONE-D-19-17279R1

Blind estimation and correction of microarray batch effect

PLOS ONE

Dear Dr. Varma,

Thank you for submitting your manuscript to PLOS ONE. After careful consideration, we feel that your manuscript may become acceptable after fixing a few remaining minor issues. Therefore, we invite you to submit a revised version of the manuscript that addresses the following 4 points:

(1) The supplemental figures are not referenced in the main text. It is up to readers to detect that there are some supplemental figures and to try to figure out what they show. So please add some sentences in the main text and explain what the supplemental figures show, like you did in your reply to the reviewers.

(2) Also, the legends of supplemental figures S5 to S7 should be improved.

For example, the current legend of Fig. S5

"S5 Fig: Consistency of calculated BES – Overlap between significant genes selected for

MSS/MSI differences in validation set 2 for BES calculated on different subsets of

the reference set."

does not reflect your reply to the comment of reviewer #1:

"I’ve included a plot of the fraction of common genes between the colon cancer/normal

dataset (validation set 2) when corrected by sets of BES calculated on the 1st + 2nd

split combined and the 3rd + 4th split combined (Supplementary Figure 5).

There is a high degree of overlap (>80% intersection) till the number of BES reaches 10.

The BES start to diverge from 20 onwards and the overlap decreases."

Similarly, the legends Figs. S6 and S7 could be improved.

For example, the legend of Fig. S6 does not explain what the black, red and blue symbols are.

The following small points refer to the line numbering of the revised manuscript with tracked changes

(3) line 108: the disadvantages ... is -> the disadvantages ... are

(4) line 387: correct plural/singular in

"We show that the batch effect between different datasets occupy a space"

We would appreciate receiving your revised manuscript by Mar 01 2020 11:59PM. To enhance the reproducibility of your results, we recommend that if applicable you deposit your laboratory protocols in protocols.io, where a protocol can be assigned its own identifier (DOI) such that it can be cited independently in the future. For instructions see: http://journals.plos.org/plosone/s/submission-guidelines#loc-laboratory-protocols

We look forward to receiving your revised manuscript.

Kind regards,

Volkhard Helms

Academic Editor

PLOS ONE

Additional Editor Comments (if provided):

The author has done a commendable job in addressing the points raised by the two reviewers.

He has performed all requested additional analysis (reviewer #1) and comparisons to other tools (reviewer #2).

The new analysis is convincing. The reply letter properly addresses all raised points.

However, when looking at the revised version of the manuscript I felt that the presentation of the additional analysis (reviewer #1) in the manuscript could be improved, see points listed above.

---

## [Author Response · Author response to Decision Letter 1]

16 Mar 2020

1) The supplemental figures are not referenced in the main text. It is up to readers to detect that there are some supplemental figures and to try to figure out what they show. So please add some sentences in the main text and explain what the supplemental figures show, like you did in your reply to the reviewers.

Response:I have added text to the manuscript describing those analyses and their results (lines 223-235 and 297-309)

2) Also, the legends of supplemental figures S5 to S7 should be improved. For example, the current legend of Fig. S5 "S5 Fig: Consistency of calculated BES – Overlap between significant genes selected for

MSS/MSI differences in validation set 2 for BES calculated on different subsets of

the reference set."does not reflect your reply to the comment of reviewer #1:

"I’ve included a plot of the fraction of common genes between the colon cancer/normal

dataset (validation set 2) when corrected by sets of BES calculated on the 1st + 2nd

split combined and the 3rd + 4th split combined (Supplementary Figure 5).

There is a high degree of overlap (>80% intersection) till the number of BES reaches 10.

The BES start to diverge from 20 onwards and the overlap decreases."

Similarly, the legends Figs. S6 and S7 could be improved.

For example, the legend of Fig. S6 does not explain what the black, red and blue symbols are.

Response: I’ve update the legends and descriptions

3) The following small points refer to the line numbering of the revised manuscript with tracked changes

a. line 108: the disadvantages ... is -> the disadvantages ... are

b. line 387: correct plural/singular in "We show that the batch effect between different datasets occupy a space"

Response: I have corrected these errors

---

## [Editor Report · Decision Letter 2]

25 Mar 2020

Blind estimation and correction of microarray batch effect

PONE-D-19-17279R2

Dear Dr. Varma,

We are pleased to inform you that your manuscript has been judged scientifically suitable for publication and will be formally accepted for publication once it complies with all outstanding technical requirements.

With kind regards,

Volkhard Helms

Guest Editor

PLOS ONE

Additional Editor Comments (optional):

The author has now properly addressed my remaining minor points.
---

## [Editor Report · Acceptance letter]

27 Mar 2020

PONE-D-19-17279R2 

Blind estimation and correction of microarray batch effect 

Dear Dr. Varma:

I am pleased to inform you that your manuscript has been deemed suitable for publication in PLOS ONE. Congratulations! Your manuscript is now with our production department. 

With kind regards,

on behalf of

Prof. Volkhard Helms 

Guest Editor

PLOS ONE